# A Link between Handgrip Strength and Executive Functioning: A Cross-Sectional Study in Older Adults with Mild Cognitive Impairment and Healthy Controls

**DOI:** 10.3390/healthcare10020230

**Published:** 2022-01-26

**Authors:** Fabian Herold, Berit K. Labott, Bernhard Grässler, Nicole Halfpaap, Corinna Langhans, Patrick Müller, Achraf Ammar, Milos Dordevic, Anita Hökelmann, Notger G. Müller

**Affiliations:** 1Department of Neurology, Medical Faculty, Otto von Guericke University, 39120 Magdeburg, Germany; berit.labott@med.ovgu.de (B.K.L.); patrick.mueller@dzne.de (P.M.); milos.dordevic@dzne.de (M.D.); notger.mueller@uni-potsdam.de (N.G.M.); 2Research Group Degenerative and Chronic Diseases, Movement, Faculty of Health Sciences Brandenburg, University of Potsdam, 14476 Potsdam, Germany; 3Institute III, Department of Sport Science, Otto von Guericke University Magdeburg, 39104 Magdeburg, Germany; bernhard.graessler@ovgu.de (B.G.); nicole.halfpaap@ovgu.de (N.H.); corinna.langhans@ovgu.de (C.L.); achraf1.ammar@ovgu.de (A.A.); anita.hoekelmann@ovgu.de (A.H.); 4Research Group Neuroprotection, German Center for Neurodegenerative Diseases (DZNE), 39120 Magdeburg, Germany; 5Center for Behavioral Brain Sciences (CBBS), 39118 Magdeburg, Germany

**Keywords:** MCI, hippocampal-prefrontal network, handgrip strength, exercise cognition, aging, brain health

## Abstract

Older adults with amnestic mild cognitive impairment (aMCI) who in addition to their memory deficits also suffer from frontal-executive dysfunctions have a higher risk of developing dementia later in their lives than older adults with aMCI without executive deficits and older adults with non-amnestic MCI (naMCI). Handgrip strength (HGS) is also correlated with the risk of cognitive decline in the elderly. Hence, the current study aimed to investigate the associations between HGS and executive functioning in individuals with aMCI, naMCI and healthy controls. Older, right-handed adults with amnestic MCI (aMCI), non-amnestic MCI (naMCI), and healthy controls (HC) conducted a handgrip strength measurement via a handheld dynamometer. Executive functions were assessed with the Trail Making Test (TMT A&B). Normalized handgrip strength (nHGS, normalized to Body Mass Index (BMI)) was calculated and its associations with executive functions (operationalized through z-scores of TMT B/A ratio) were investigated through partial correlation analyses (i.e., accounting for age, sex, and severity of depressive symptoms). A positive and low-to-moderate correlation between right nHGS (rp (22) = 0.364; *p* = 0.063) and left nHGS (rp (22) = 0.420; *p* = 0.037) and executive functioning in older adults with aMCI but not in naMCI or HC was observed. Our results suggest that higher levels of nHGS are linked to better executive functioning in aMCI but not naMCI and HC. This relationship is perhaps driven by alterations in the integrity of the hippocampal-prefrontal network occurring in older adults with aMCI. Further research is needed to provide empirical evidence for this assumption.

## 1. Introduction

Handgrip strength was found to be an important marker of health in general [1,2,3,4] and brain health in particular [5,6,7]. Indeed, a stronger handgrip has been linked to superior cognitive performance in younger adults [8], in middle-aged adults [9], and in older adults [10,11,12,13,14,15,16,17,18]. Furthermore, in older adults, higher levels of handgrip strength are associated with lesser cognitive decline during aging [19,20,21,22,23,24,25,26]. In sum, these studies suggest that a relationship between measures of handgrip strength and cognitive performance exists. Hence, handgrip strength may be a clinically useful marker to identify individuals at high risk of developing mild cognitive impairment (MCI) [27,28], and/or dementia (e.g., Alzheimer’s Disease) [29,30].

However, the majority of the mentioned studies used rather global measures of cognitive performance (e.g., Mini-Mental State Examination (MMSE)) [19,20,22,24,25,26] while the empirical evidence regarding the association of handgrip strength with specific cognitive domains (e.g., executive functions) is not exhaustive. Interestingly, there is some evidence that executive functions, especially task switching and cognitive flexibility, are compromised in older individuals with amnestic MCI (aMCI) [31,32,33]. Importantly, older adults with aMCI who in addition to their memory deficits suffer from a frontal-executive dysfunction have a higher risk of developing dementia compared to older adults with aMCI with additional visuospatial or language dysfunction [34].

To allow a timely initiation of interventions aiming to lower the burdens of neurological disorders (e.g., dementia), an early identification of adults being at a high risk of developing them (e.g., older adults with aMCI and executive dysfunction) is mandatory. Notably, in the literature, the hypothesis that motoric measures (e.g., handgrip strength) and higher-order cognitive functions (e.g., executive functions) share a set of common neural substrates (e.g., frontal cortex, hippocampus) was proposed [5]. Accordingly, motoric measures (e.g., handgrip strength) might be a valuable and easily applicable parameter to identify adults at higher risk of developing neurological disorders such as MCI [27,28] and/or dementia [29,30]. In this context, and with regard to the idea of shared neural substrates, there is evidence in the literature (i) that the hippocampus is involved in memory and executive functions in adults [35,36,37] and (ii) that handgrip strength is related to the (right) hippocampal volume in healthy adults and in adults with a major depressive disorder [38]. These findings suggest that the hippocampus could be, among other brain structures such as the frontal cortex, a neural substrate that is shared by higher-order cognitive functions (i.e., executive functions) and motoric measures (i.e., handgrip strength). Moreover, there is evidence that the hippocampal volume is influenced by the subtype of MCI as it was observed that older adults with aMCI have a lower hippocampal volume as compared to older adults with naMCI [39]. Whether such a difference in the shared neural substrate (e.g., hippocampal volume) is also mirrored in behavioral performance (i.e., the relationship between measures of executive functions and handgrip strength) has not been extensively studied. Again, an early recognition of adults being at high risk of developing dementia (e.g., older adults with aMCI and executive dysfunction) is essential to initiate appropriate interventions, and thus the investigation of possible relationships between measures of handgrip strength and executive functioning in older adults with different subtypes of MCI is of great practical relevance. Hence, this study aims to investigate the relationships between measures of handgrip strength (i.e., assessed with a handheld dynamometer) and executive functioning (i.e., operationalized as performance in Trail Making Test (TMT)) in older adults with different subtypes of MCI (amnestic and non-amnestic) and healthy older adults. Based on the available evidence [10,11,12,13,14,15,16,19,20,21,22,23,24,25], we hypothesize that positive correlations between measures of HGS and executive functioning exist, and that the magnitude of the associations might be a function of the cognitive status of the older adults.

## 2. Materials and Methods

### 2.1. Participants

In the current study, older adults with aMCI, naMCI, and healthy controls (HC) were recruited as part of a larger project (MyFit study [40]) through advertisements in local newspapers, flyers, posters, word of mouth, and by using existing databases. After recruitment, the individuals were screened for eligibility based on the following inclusion criteria: (i) 50 to 80 years old, (ii) native German-speaking, and (iii) able to manage everyday activities independently. Individuals who had poor or uncorrected vision/hearing or color weakness/blindness, and/or suffer from (a) severe psychiatric disorders (e.g., bipolar disorder) or depression (assessed via the Geriatric Depression Scale (GDS; 15 items; cut-off score ≥ 6) [41]), (b) severe orthopaedic diseases (e.g., a bone fracture in last six months, herniated vertebral disc), (c) severe muscular diseases (e.g., myositis, tendovaginitis), (d) severe cardiovascular diseases (e.g., heart insufficiency), (e) severe endocrinologic diseases (e.g., manifest hypothyroidism or hyperthyroidism, insulin dependent diabetes mellitus type II, BMI > 30), (f) neurological diseases other than MCI (e.g., stroke, epilepsy, Multiple Sclerosis), (g) major injury or had major surgery in the last six months, and/or use neuroleptics, narcotic analgesics, benzodiazepines, or psychoactive medications, and/or consume illegal intoxicants and/or have an alcohol abuse, and (h) are pregnant were excluded.

As shown in Figure 1, three participants of the MCI groups and three participants of the HC group were excluded from the analysis due to the severity of depressive symptoms. Due to diseases of the hands, three participants from the MCI group were also excluded. Furthermore, only right-handers were included in the analysis who were determined by the short version (10 items) of the Edinburgh Handedness Inventory [42] (EHI; cut-off score ≥50 indicated right-handedness; <50 to >−50 indicate ambidextrous handedness; ≤−50 indicated left-handedness [43]). Hence, due to left-handedness, six participants in the MCI groups and four participants in the HC group were excluded from the analysis (see Figure 1).

To identify older adults with MCI the general recommendations of Winblad et al. [44] and Peterson et al. [45], who define MCI as cognitive performance below the age-appropriate level without symptoms indicating the presence of manifest dementia (DSM IV, ICD 10) were followed. Accordingly, individuals with MCI were characterized by (i) the preservation of basic activities of daily living and minimal impairment in complex instrumental functions, (ii) a self and/or informant-reported impairment on objective cognitive tasks, and/or (iii) evidence of decline over time on objective cognitive tasks [44]. To screen for these criteria, the Alzheimer’s Disease Consortium to Establish a Registry (CEARD) Plus test battery was used [46] and mild cognitive impairment was operationalized by an under average performance (i.e., a performance below −1.5 SD in the age, sex, and education-adjusted z-value) in at least one subtest of the CEARD Plus test battery while minimum performance in the MMSE as part of the CERAD was set at 24 and above (to exclude cases of dementia) [47]. As recommended [44], participants meeting these criteria were referred to an experienced neurologist who verified (or refuted) the diagnosis of MCI. Furthermore, the neurologist performed a standard clinical examination thereby ensuring that handgrip strength was not influenced by diseases such as polyneuropathy or myopathy. Participants not meeting the clinical criteria of MCI were allocated to the HC group if they wanted to participate further (see Figure 1).

The performance (i.e., saving score) of the delayed recall trial of the Wordlist and Figure episodic memory test (included in the CERAD test battery) were used as criteria to differentiate between aMCI and naMCI. In accordance with the recommendations in the literature, participants with a performance below −1.5 SD in the age, sex, and education-adjusted z-value in at least one of these two cognitive subtests of the CERAD test battery were classified as aMCI [48]. Please note that this study is part of a larger project (MyFit study [40]) and in the current study we performed the analysis of selected and secondary outcome measures of this larger trial. Thus, no additional sample size calculation was performed as the available data of participants who had been recruited for the MyFit study were used (see reference [40] for more detailed information and sample size calculation of the MyFit study).

### 2.2. Assessment of Cognitive Performance and Handgrip Strength

As the data presented in this study were collected in the context of a larger project (MyFit project [40]), the participants were asked to visit our laboratory several times as described elsewhere [40]. In this study, the parameters of interest were executive functioning (assessed via Trail Making Test) and handgrip strength (assessed with a handheld dynamometer (Trailite TL-LSC100, LiteXpress GmbH, Ahaus, Germany), and thus only these measures are reported in more detail.

The Trail Making Test (TMT A and B) is part of the CERAD Plus test battery. It was conducted as described in [49]. While the TMT A probes visual search performance, the TMT B assesses cognitive flexibility [50,51,52]. In TMT A, a series of 25 encircled numbers has to be connected in ascending order [50,53]. In TMT B, 25 encircled numbers and letters have to be connected in an alternating and ascending order (e.g., 1 with A, then 2, then B) [50,53]. With respect to the aim of our study, the performance measures of TMT A and TMT B (time needed to complete the task) were used to calculate the ratio (TMT B/A) as a measure of the individual shifting ability [54,55]. This ratio was proposed to reflect executive functioning better than other performance measures obtained via the TMT (e.g., time to complete TMT A) [49,56].

Maximal handgrip strength was assessed based on the Southampton protocol [57]. The participants (i) were seated in a comfortable chair with their feet flat on the ground, (ii) were advised to adduct their shoulders and remain them neutrally rotated, (iii) were asked to flex the elbow of the tested extremity at 90° while maintaining a neutral wrist position (i.e., thumb facing upward), and (iv) were asked to squeeze the hand as hard as they could for three seconds to assess their handgrip strength [57]. Each participant conducted three trials for each hand and was asked to change the hand after performing one trial [58,59,60]. The best trial (i.e., the trial with the highest absolute handgrip strength) of three trials of each extremity side was used to calculate the normalized handgrip strength. Maximal handgrip strength was normalized to the body mass index (BMI) of the participants to account for the influence of anthropometric factors (e.g., body mass and body height) as carried out in [58,59]. The normalized handgrip strength (nHGS) for each hand was calculated as follows: normalized handgrip strength (nHGS) = absolute handgrip strength (in kg)/BMI (in kg/m^2^) [58,59,61]. The nHGS was used for further statistical analysis.

Prior to the assessment, participants were briefed about the experimental procedure and informed of possible risks and benefits associated with the study. All participants provided written consent to participate and received financial compensation. All study procedures were in accordance with the latest version of the Declaration of Helsinki, had been approved by the local Ethics Committee of the Medical Faculty of the Otto von Guericke University Magdeburg (reference number: 83/19) and were pre-registered in ClinicalTrials.gov (NCT04427436 on the 10 June 2020).

### 2.3. Statistical Analysis

The statistical analysis was performed using JAMOVI (version 2.2.2 current) [62]. Non-parametric tests (i.e., Kruskal–Wallis and Dwass–Steel–Critchlow–Fligner (as post-hoc tests)) were applied to compare MCI groups and HC group concerning age, body height, body mass, BMI, educational level, GDS, nHGS (left and right), and TMT performance (operationalized through z-score of TMT B/A ratio). With regard to the Kruskal–Wallis test, epsilon square (ε^2^) was calculated as a measure of effect size and rated as follows: ≥0.01 to <0.6: small effect; ≥0.06 to <0.14: medium effect; ≥0.14: large effect [63].

Non-parametric partial correlation coefficients (i.e., Spearman’s rho [r_p_]; accounting for age, sex, and severity of depressive symptoms) were calculated to investigate possible relationships between executive functions and nHGS. Furthermore, the GDS score was used as a covariate since a relationship between measures of HGS and severity of depressive symptoms was reported in older adults [64]. Based on previous studies [10,11,12,13,14,15,16,19,20,21,22,23,24,25], one-tailed significance tests were used for the correlational analyses as a positive relationship between nHGS and executive functioning is assumed. The partial correlation coefficient r_p_ was rated as follows: <0.19: no correlation; ≥0.20 to ≤0.39: low correlation; ≥0.40 to ≤0.59: moderate correlation; ≥0.60 to ≤0.79: moderately high correlation; ≥0.8: high correlation [65,66].

In addition, the cocor package (one-tailed significance test) was used to compare the correlation coefficients between older adults with aMCI, naMCI, and HC [67].

For all statistical tests, the level of significance was set to α = 0.05.

## 3. Results

The general characteristics of the participants are shown in Table 1 (see also Appendix A for further information). We observed a significant effect of group concerning body height (χ^2^ (df = 2; *n* _aMCI_ = 22, *n* _naMCI_ = 21, *n* _HC_ = 27) = 10.547, *p* = 0.005, ε^2^ = 0.153), body mass (χ^2^ (df = 2; *n* _aMCI_ = 22, *n* _naMCI_ = 21, *n*
_HC_ = 27) = 6.446, *p* = 0.040, ε^2^ = 0.093), GDS (χ^2^ (df = 2; *n* _aMCI_ = 22, *n* _aMCI_ = 21, *n* _HC_ = 27) = 6.975, *p* = 0.031, ε^2^ = 0.101) and MMSE (χ^2^ (df = 2; *n* _aMCI_ = 22, *n* _aMCI_ = 21, *n* _HC_ = 27) = 23.700, *p* ≤ 0.001, ε^2^ = 0.343). The post-hoc tests concerning body height show that aMCI (W (*n*
_aMCI_= 22, *n*
_HC_ = 27) = −3.630, *p* = 0.028) and naMCI (W (*n* _naMCI_ = 21, *n* _HC_ = 27) = −4.063, *p* = 0.011) were taller than HC. However, the post-hoc tests concerning the comparison of the groups with respect to body mass did not reach statistical significance although the difference between naMCI and HC was marginally non-significant (W (*n* _naMCI_ = 21, *n* _HC_ = 27) = −3.310, *p* = 0.051).

With respect to GDS, the post-hoc tests show that the GDS score was higher in naMCI as compared to HC (W (*n* _naMCI_ = 21, *n* _HC_ = 27) = −3.780, *p* = 0.021). The post-hoc test concerning MMSE score revealed that aMCI (W (*n*
_aMCI_ = 22, *n*
_HC_ = 27) = 6.198, *p* ≤ 0.001) and naMCI (W (*n* _naMCI_ = 21, *n* _HC_ = 27) = 5.457, *p* ≤ 0.001) performed worse than HC but there was no difference between aMCI and naMCI (W (*n*
_aMCI_ = 22, *n* _naMCI_ = 21) = −0.834, *p* = 0.826). No other between-group comparison was statistically significant (i.e., age, BMI, educational level, nHGS of left and right hand, and z-score of TMT B/A).

We observed a positive low-to-moderate correlation between left nHGS (r_p_ (22) = 0.420, *p* = 0.037) and right nHGS (r_p_ (22) = 0.364, *p* = 0.063) with executive functioning (i.e., operationalized via z-score of TMA B/A ratio) in the aMCI group (see Figure 2a). In addition, in aMCI, we did not observe a significant difference between the correlation coefficients of the left hand (i.e., reaching statistical significance) and the right hand (i.e., slightly missed to reach statistical significance) with executive functioning (*p* > 0.05) using the cocor package [67].

As shown in Figure 2b,c, no significant correlations were observed between nHGS and executive functioning in the naMCI group (left nHGS (r_p_ (20) = 0.004, *p* = 0.494); right nHGS (r_p_ (21) = 0.007, *p* = 0.489)) and in the HC group (left nHGS (r_p_ (27) = 0.007; *p* = 0.486); right nHGS (r_p_ (27) = −0.011, *p* = 1.000)). Furthermore, there were no between-group differences with respect to the comparison of aMCI and naMCI concerning the correlation coefficients of left nHGS (z = 1.329, *p* = 0.092) and right nHGS (z = 1.139, *p* = 0.127) with executive functioning. Furthermore, no group differences between aMCI and HC were observed regarding the correlation coefficients of left nHGS (z = 1.435, *p* = 0.076) and right nHGS (z = 1.278, *p* = 0.101) with executive functioning. In addition, there were also no between-group differences with respect to the comparison of naMCI and HC concerning the correlation coefficients of left nHGS (z = −0.010, *p* = 0.496) and right nHGS (z = 0.058, *p* = 0.477) with executive functioning.

## 4. Discussion

This study investigated possible links between measures of handgrip strength and executive functioning in older adults with different subtypes of MCI and HC. We observed that in aMCI, stronger nHGS was associated with better performance in executive functions (operationalized by the z-score of the TMT B/A ratio) but not in naMCI and/or in HC, although there were no between-group differences concerning the correlation coefficients. This observation is, at least partly, in line with previous studies reporting a comparable relationship between cognitive performance and handgrip strength in older adults with diabetes [68] and in older adults with MCI and dementia [69]. However, in the study of Hesseberg et al. [69], MCI patients were not further classified into amnestic or non-amnestic subtypes. Given that there are significant differences between older adults suffering from aMCI and naMCI (i) with respect to the conversion rates to dementia (i.e., conversion rate to AD is higher in aMCI as compared to naMCI [70,71]), (ii) with respect to brain changes (e.g., lower cortical thickness in entorhinal cortex, the fusiform gyrus, the precuneus and the isthmus of the cingulate gyrus [72] and hippocampal volume [39] in aMCI as compared to naMCI), and (iii) with respect to motoric measures (e.g., slower gait speed, especially in dual-task conditions, in aMCI as compared to naMCI [73]), a differentiation between different subtypes of MCI, as performed in this study, is favorable.

The absence of a correlation in the naMCI and HC group in the present study is perhaps related to the fact that in these groups the neural substrates that are important for handgrip strength and executive functioning (i.e., operationalized by TMT performance) are better preserved. In the literature, there is considerable evidence that executive functions in general [74,75], and the execution of TMT in particular [76,77,78,79,80,81], rely on the integrity of the prefrontal cortex (PFC). Accordingly, the absence of a correlation in the naMCI and HC groups might reflect that the PFC is (more) intact in these two groups.

Following this line of interpretation, the moderate and positive correlation in the aMCI group might indicate that integrity of the PFC is compromised in individuals with a relatively low handgrip strength whereas, vice versa, a relative high handgrip strength signifies better integrity of the PFC in aMCI. In line with this assumption, there is evidence that higher handgrip strength is linked to (i) more pronounced task-related cortical hemodynamics in the PFC in younger adults [60] and (ii) superior white matter integrity in the frontal cortex in older adults [82]. These findings suggest that higher handgrip strength is linked to better integrity of the frontal cortex although future research is needed to buttress this assumption empirically [60,83]. Notably, there is evidence that the integrity of the frontal cortex is compromised in individuals with aMCI encompassing (i) alterations in task-related PFC activation [84,85,86], (ii) changes in gray matter integrity in frontal brain areas [87], and (iii) cortical thinning in frontal brain areas [34] in older adults with aMCI compared to HC. Moreover, it was observed that brain alterations in the frontal cortex (e.g., in white matter) in older adults with MCI correlate with performance changes in executive functioning (e.g., TMT B) [88]. Thus, a higher nHGS in older adults with aMCI might reflect better preservation of these neural correlates (e.g., frontal cortex) in these individuals which, in turn, allows for better performance in tasks probing executive functioning (e.g., TMT B). This assumption nicely fits with the idea that handgrip strength shares specific neural correlates with higher-order cognitive functions (e.g., of the frontal cortex) [5] and with the evidence showing that in older adults with aMCI motoric measures (i.e., gait speed) are correlated with the grey matter volume of frontal cortical regions [89]. In this context, it seems reasonable to speculate that such associations are not only driven by changes in the PFC but rather by alterations of a complex hippocampal-prefrontal network given the evidence that the hippocampus is involved in executive functioning in adults [35,36,37] and that handgrip strength is related to the (right) hippocampal volume in healthy adults and in adults with a major depressive disorder [38]. Vice versa, there is also evidence that the PFC is involved in memory processes (for review see [90,91]), which supports the idea that the relationship between measures of handgrip strength and executive functioning becomes evident especially in individuals with impaired memory function such as older adults with aMCI. However, future research is warranted to confirm this assumption empirically by applying neuroimaging techniques [60,83].

Given that individuals with aMCI, and especially those with executive dysfunctions [34,92,93], have a relatively high risk of developing AD [70,71,94,95,96], the difference in the association of nHGS and executive functioning between aMCI and naMCI could be of high clinical relevance (even if the comparison of those correlations did not reach statistical significance in this study), as it suggests that in individuals with aMCI a relatively high level of (handgrip) strength can indicate preserved executive functions and, therefore, a lower risk of conversion to dementia. Of course, the latter assumption needs to be verified in a long-term study.

In addition, we observed that the correlation between nHGS and executive functioning reached statistical significance concerning the left hand but not concerning the right hand (see also Figure 2a). However, given the finding (i) that there is no statistically significant difference between both correlation coefficients (*p* > 0.05) and (ii) that the correlation between nHGS of the right hand with executive functioning was close to reaching statistical significance (*p* = 0.063), this finding should not be overinterpreted. Although this cross-sectional study does not allow us to elucidate causal relationships underlying the association of handgrip strength and cognitive performance (i.e., executive functioning), our finding fits with the available evidence suggesting that resistance training can be a beneficial intervention strategy to improve brain structure and function in both healthy adults [83,97,98,99,100] and in older adults with MCI [101,102,103]. Based on our findings, future research should investigate whether older adults with different subtypes of MCI (e.g., aMCI vs. naMCI) would benefit differently from resistance training interventions.

In summary, our findings suggest that in older adults with aMCI, higher levels of nHGS are associated with a better performance in executive functioning. This relationship is possibly caused by alterations in brain networks that accompany aMCI such as the PFC (e.g., hippocampal-prefrontal network). However, to confirm these assumptions future studies are needed that investigate the associations between measures of handgrip strength, cognitive performance (e.g., executive functions), and their neural correlates (e.g., functional cortical hemodynamic changes in the PFC) [60,83].

## 5. Limitations

There are some limitations of the current study that need to be acknowledged. Firstly, the sample size was relatively small and only right-handed individuals were included in the analysis. Secondly, not all core components of executive functioning (e.g., inhibition and working memory) were assessed. Thirdly, no multiple comparison adjustments were performed. In this context, there is an ongoing discussion about when and how it is necessary to adjust for multiple comparisons [104,105,106] and it is stated that in exploratory studies, multiple comparison adjustments are not strictly required [105]. Hence, our findings should be interpreted cautiously, and further research with larger sample sizes is needed in order to confirm (or refute) our findings. Furthermore, additional research that considers changes on multiple levels of analysis (e.g., changes on molecular and cellular level, changes on functional and structural brain level, and socioemotional changes) is also necessary to deepen our knowledge about neurobiological mechanisms driving the relationships between measures of handgrip strength and cognitive performance [83].

## 6. Conclusions

The findings of the current study suggest that higher levels of nHGS are related to better executive functioning in aMCI but not in naMCI and in HC. Based on the available evidence, we hypothesize that this relationship may be driven by alterations in the integrity of the hippocampal-prefrontal network occurring in older adults with aMCI. However, further research is needed to provide direct empirical evidence for this assumption.

## Figures and Tables

**Figure 1 healthcare-10-00230-f001:**
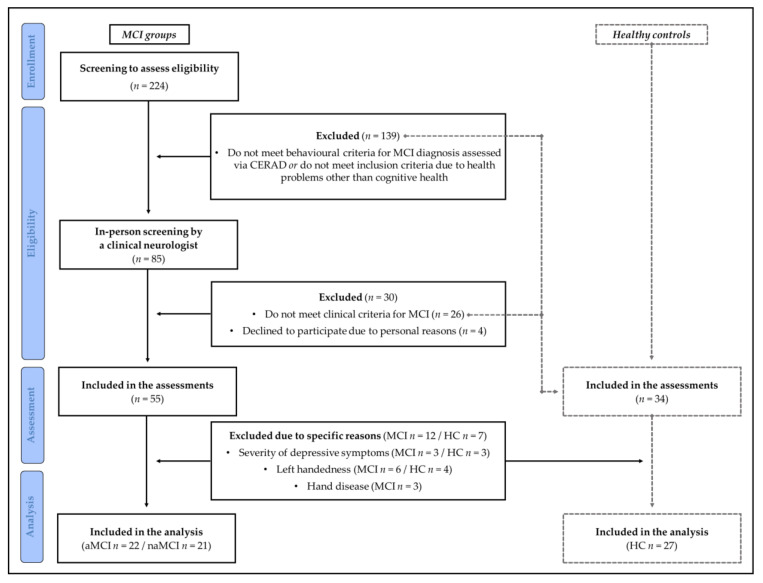
Flow diagram that schematically depicts the selection processes (from screening to assess eligibility to the final inclusion in the statistical analysis) and the reasons for exclusion. aMCI: amnestic mild cognitive impairment; CERAD: Alzheimer’s Disease Consortium to Establish a Registry test battery; HC: healthy controls; naMCI: non-amnestic mild cognitive impairment.

**Figure 2 healthcare-10-00230-f002:**
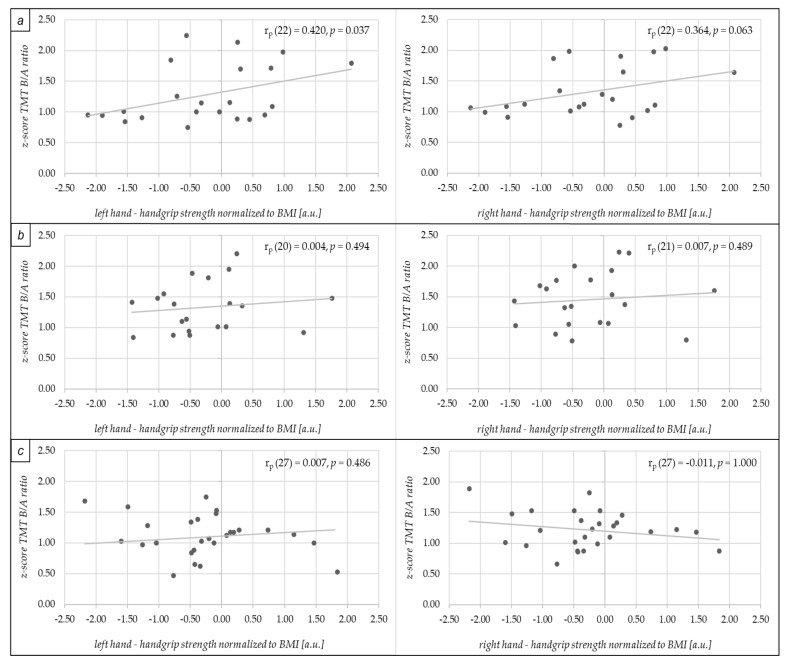
Scatter plots displaying the correlations between normalized handgrip strength (nHGS) of the left hand and right hand and z-scores of TMT B/A ratio (reflecting executive functioning) in the older individuals with amnestic mild cognitive impairment (aMCI) in (**a**), for older adults with non-amnestic mild cognitive impairment (naMCI) in (**b**) and healthy older controls (HC) in (**c**). The handgrip strength was normalized to the Body Mass Index to account for the influence of anthropometrics on handgrip strength [58,59]. a.u.: arbitrary unit; BMI: Body Mass Index; r_p_: partial correlation coefficient (accounting for age, sex, and severity of depressive symptoms [via scores in Geriatric Depression Scale]); TMT: Trail Making Test.

**Table 1 healthcare-10-00230-t001:** Overview of the general characteristics of the participants.

General Characteristics of the Participants	Median ± Interquartile Range (Minimum to Maximum)		
	aMCI (*n* = 22)	naMCI (*n* = 21)	HC (*n* = 27)
**Female/Male (n)**	14/8	9/12	19/8
**Age (years)**	69 ± 9 (60 to 81)	71 ± 8 (56 to 80)	68 ± 10 (54 to 83)
**Body height (cm)**	171.0 ± 11.0 * (150.0 to 184.0)	173.0 ± 13.0 ^#^ (159.0 to 189.0)	165.0 ± 9.5 (156.0 to 179.0)
**Body mass (kg)**	72.0 ± 15.0 (61.0 to 93.0)	77.0 ± 8.0 (54.4 to 94.4)	67.0 ± 22.5 (50.0 to 94.0)
**BMI (kg/m^2^)**	24.1 ± 4.2 (20.9 to 29.1)	25.8 ± 1.6 (21.4 to 28.5)	24.7 ± 5.9 (19.3 to 31.0)
**Educational level (years)**	15 ± 4 (11 to 20)	15 ± 3 (11 to 18)	15 ± 3 (12 to 18)
**GDS (total score)**	1.5 ± 3.0 (0.0 to 4.0)	2.0 ± 2.0 ^#^ (0.0 to 5.0)	1.0 ± 1.5 (0.0 to 3.0)
**EHI (score)**	100.0 ± 23.3 (52.9 to 100.0)	100.0 ± 0.0 (73.3 to 100.0)	100.0 ± 21.1 (53.9 to 100.0)
**nHGS left/right (a.u.)**	1.05 ± 0.76/1.12 ± 0.62/ (0.75 to 2.24/0.78 to 2.03)	1.37 ± 0.51 ^a^/1.43 ± 0.70 (0.84 to 2.20/0.79 to 2.23)	1.12 ± 0.33/1.21 ± 0.42 (0.47 to 1.75/0.66 to 1.89)
**TMT B/A (z-score)**	−0.18 ± 1.20 (−2.13 to 2.07))	−0.47 ± 0.89 (−1.43 to 1.76)	−0.32 ± 0.74 (−2.18 to 1.84)
**MMSE (points)**	27.0 ± 1.8 * (25.0 to 30.0)	27.0 ± 2.0 ^#^ (24.0 to 30.0)	28.0 ± 1.0 (27.0 to 30.0)

^a^ Please note that values of nHGS left in the naMCI group were based on *n* = 20 since the data of one participant in the naMCI was not used to calculate median, interquartile range and minimum to maximum due to drop hand symptomatic in the left (non-dominant) hand. * indicates significant differences between aMCI and HC; **^#^** indicates significant differences between naMCI and HC. a.u.: arbitrary unit; BMI: Body Mass Index; EHI: Edinburgh Handedness Inventory (cut-off score ≥ 50 indicated right-handedness; <50 to >−50 indicate ambidextrous handedness; ≤−50 indicated left-handedness [43]); GDS: Geriatric Depression Scale (cut-off score ≥ 6 [41]); MMSE: Minimal Mental State Examination; nHGS: normalized handgrip strength; TMT: Trail Making Test.

## Data Availability

The data presented in this study are available on request from the corresponding author. The data are not publicly available in order to protect participants’ privacy.

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
