# Peer review of "A Link between Handgrip Strength and Executive Functioning: A Cross-Sectional Study in Older Adults with Mild Cognitive Impairment and Healthy Controls"

_healthcare, 2022, doi:10.3390/healthcare10020230_

Round 1

Reviewer 1 Report

The manuscript is about potential associations between handgrip strength and executive functioning in aMCI, naMCI, and healthy older adults. The results are in favor of the existence of such an association only in aMCI. The manuscript is well-written, and it is an interesting finding for future studies and clinical applications. I have a few points that I’d like the authors to address:

  • Fig 1 has low quality and can be improved
  • Fig 2 can become more informative. It currently presents the results only for aMCI. Please add the figures for naMCI and healthy groups as well, so that the correlations could also be compared visually. In addition, please add the actual BMI as color codes.
  • Please conduct a linear regression analysis and report which one of the basic characteristics including age and sex mentioned in Table 1 could explain actual handgrip strength better for each of the three groups.
  • The discussions about neural networks (l. 272-302) are too details for the scope of this study. This is not a brain imaging study and thus weak detailed assumptions should be avoided. Similarly for l.311-317.
  • The HGS was measured for the left and right hands but it is not sufficiently discussed. For example, why there is a significant correlation in left hand but not right hand as shown in fig 2.
  • More information about the adjustment for multiple comparisons should be provided. Please provide transparent explanation at least as a supplementary result on what changes if you apply the adjustments. There are different adjustment methods, and some are less conservative than others that might maintain significance, otherwise effect size would also be an important measure to report and discuss.

I would suggest a revision of this manuscript to address these issues.

Author Response

The manuscript is about potential associations between handgrip strength and executive functioning in aMCI, naMCI, and healthy older adults. The results are in favor of the existence of such an association only in aMCI. The manuscript is well-written, and it is an interesting finding for future studies and clinical applications. I have a few points that I’d like the authors to address:

Fig 1 has low quality and can be improved

  • We thank the reviewer for this hint. We enlarged the figure and hope that the readability is now better.

Fig 2 can become more informative. It currently presents the results only for aMCI. Please add the figures for naMCI and healthy groups as well, so that the correlations could also be compared visually. In addition, please add the actual BMI as color codes.

  • We are grateful for the constructive feedback. As suggested by the reviewer, we have added the Figures for naMCI and healthy controls. However, we did not add color codes for BMI as the there is no strong dispersion of BMI values in our cohorts (see Table 1 and exclusion criteria of overweight individuals – see line 111 to 113 in the revised version of the manuscript) “…(e) severe endocrinologic diseases (e.g., manifest hypothyroidism or hyperthyroidism, insulin dependent diabetes mellitus type II, BMI > 30),...”]”.

Please conduct a linear regression analysis and report which one of the basic characteristics including age and sex mentioned in Table 1 could explain actual handgrip strength better for each of the three groups.

  • We thank the reviewer for her/his advice. Although a linear regression is undoubtedly helpful to elucidate which factor can explain actual handgrip strength, we are somewhat reluctant to perform this type of statistical analysis given the following facts: (i) To perform a linear regression, normal distribution is required. However, not all parameters have normal distribution and thus the requirement for the application of a linear regression is violated which, in turn, might produce spurious results. (ii) Another issue related to linear regression analysis is our limited sample size. Albeit it is still an open issue what constitutes the minimum sample size for a linear regression analysis (Green, 1991; Jenkins & Quintana-Ascencio, 2020; Maxwell, 2000), as a rule of thumb at least 10 observations per variable (predictor) should be available to perform this type of statistical analysis. Thus, adding all the basic characteristics displayed in Table 1 (as suggested by the reviewer) is somewhat difficult and might lead to spurious results. Based on these two facts, we refrain from performing a linear regression analysis, although we think that this could be a promising idea for further studies with a larger sample size.

The discussions about neural networks (l. 272-302) are too details for the scope of this study. This is not a brain imaging study and thus weak detailed assumptions should be avoided. Similarly for l.311-317.

  • We are thankful for the reviewer’s feedback. However, we believe that a discussion of potentially underlying neural networks is essential for a scientific paper and given that the discussion is already relatively short (around one and a half pages), we refrained from a more vigorous shortening.

The HGS was measured for the left and right hands but it is not sufficiently discussed. For example, why there is a significant correlation in left hand but not right hand as shown in fig 2.

  • We thank the reviewer for this important hint. To address this issue, we run an additional analysis and added the following to the results section (see line 256 to 259 in the revised version of the manuscript): “In addition, in aMCI we did not observe a significant difference between the correlation coefficients of the left hand (i.e., reaching statistical significance) and the right hand (i.e., slightly missed to reach statistical significance) with executive functioning (p > 0.05) using the cocor package [67].”
  • Furthermore, we added the following to our discussion section (line 350 to 355 in the revised version of the manuscript): “In addition, we noticed that the correlation between nHGS and executive functioning reached statistical significance concerning the left hand but not concerning the right hand (see also Figure 2 a). However, given the finding (i) that there is no statistically significant difference between both correlation coefficients (p > 0.05) and (ii) that the correlation between nHGS of the right hand with executive functioning was close to statistical significance (p = 0.063), this finding should not be overinterpreted.”

More information about the adjustment for multiple comparisons should be provided. Please provide transparent explanation at least as a supplementary result on what changes if you apply the adjustments. There are different adjustment methods, and some are less conservative than others that might maintain significance, otherwise effect size would also be an important measure to report and discuss.

  • We agree with the reviewer that statistical adjustment is a critical point. We had already stated the following in the manuscript to reflect this point: “Thirdly, no multiple comparison adjustments were performed for the statistical analysis. In this context, there is an ongoing discussion about when and how it is necessary to adjust for multiple comparisons [103–105] and it is stated that in exploratory studies, multiple comparison adjustments are not strictly required [104]. Hence, our findings should be interpreted cautiously, and further research with larger sample sizes is needed to confirm (or refute) in order to generalize our findings.“
  • We also agree with the reviewer that effect sizes are important statistical measures. We already have provided effect sizes for our interferential statistical comparisons (see line 223 to 227 in the revised version of the manuscript). 

References

American Psychological Association. (2020). Publication manual of the American Psychological Association: The official guide to APA style (Seventh edition). American Psychological Association. https://doi.org/10.1037/0000165-000

Green, S. B. (1991). How Many Subjects Does It Take To Do A Regression Analysis. Multivariate Behavioral Research, 26(3), 499–510. https://doi.org/10.1207/s15327906mbr2603_7

Jenkins, D. G., & Quintana-Ascencio, P. F. (2020). A solution to minimum sample size for regressions. PLOS ONE, 15(2), e0229345. https://doi.org/10.1371/journal.pone.0229345

Maxwell, S. E. (2000). Sample size and multiple regression analysis. Psychological Methods, 5(4), 434–458. https://doi.org/10.1037/1082-989x.5.4.434

Reviewer 2 Report

Dear authors, 

The manuscript is interesting, but requires minor revisions. I also suggest improving the flow. Sometimes the manuscript is difficulto to follow due to many abbreviations and excessive information. Reduce to minimal necessary.

My specific comments are attached.

Regards.

Author Response

Dear authors,

The manuscript is interesting, but requires minor revisions. I also suggest improving the flow. Sometimes the manuscript is difficult to follow due to many abbreviations and excessive information. Reduce to minimal necessary.

Line 64 - All the information regarding the rationale of the study must be inserted in the introduction.

  • We thank the reviewer for this hint. We revised the section as follows (see line 64 to 91 in the revised version of the manuscript): “To allow for a timely onset of interventions aiming to lower the burdens of neurological disorders (e.g., dementia), an early identification of adults being at a high risk to develop them (e.g., older adults with aMCI and executive dysfunction) is mandatory. Notably, in the literature the hypothesis that motoric measures (e.g., handgrip strength) and higher-order cognitive functions (e.g., executive functions) share a set of common neural substrates (e.g., frontal cortex, hippocampus) has been proposed [5]. Accordingly, motoric measures (e.g., handgrip strength) might be a valuable and easy applicable parameter to identify adults at higher risk to develop neurological disorder such as MCI [27,28] and/or dementia [29,30]. In this context and with regard to idea of shared neural substrates, there is the evidence in the literature (i) that the hippocampus is involved in memory and executive functions in adults [35–37] and (ii) that that handgrip strength is related to the (right) hippocampal volume in healthy adults and in adults with a major depressive disorder [38]. These findings suggest that the hippocampus could be, among other brain structures such as the frontal cortex, a neural substrate that is shared by higher-order cognitive functions (i.e., executive functions) and motoric measures (i.e., handgrip strength). Moreover, there is evidence that the hippocampal volume is influenced by the subtype of MCI as it has been observed that older adults with aMCI have a lower hippocampal volume as compared to older adults with naMCI [39]. Whether such a difference in shared neural substrate (e.g., hippocampal volume) is also mirrored in behavioral performance (i.e., relationship between measures of executive functions and handgrip strength) has not been extensively studied. Again, as an early identification of adults being at high risk to develop dementia (e.g., older adults with aMCI and executive dysfunction) is mandatory for a timely onset of interventions, the investigation of possible relationships between measures of handgrip strength and executive functioning in older adults with different subtypes of MCI is of great practical relevance.”

Line 74 - Sample size calculation is necessary in this section.

  • We are grateful for the reviewer’s feedback. However, as this study analyzed secondary outcome measures of a larger trial, no additional sample size calculation was performed. To clarify this, the following has been added to the manuscript (see line 157 to 162 in the revised version of the manuscript): “Please note that this study is part of a larger project (MyFit study [40]) and in the current study we performed the analysis of selected and secondary outcome measures of this larger trial. Thus, no additional sample size calculation was performed as the available data of participants who had been recruited for the MyFit study was used (see reference [40] for more detailed information and sample size calculation of the MyFit study).”

Line 121/122 - Is this the gold standard? Please, clarify.

  • We are thankful to pointing out this issue. Currently, there is no established gold standard available how to diagnose MCI. Thus, we follow best practice recommendations (Winblad et al., 2004). To acknowledge this, we have added the following to the revised version of our manuscript (line 146 to 148): “As recommended [44], participants meeting these criteria were referred to an experienced neurologist who verified (or refuted) the diagnosis of MCI.”

Line 135/136 - Again, all the rationale about the study must be presented. If you need, address as a supplemental material.

  • We thank the reviewer for her/his valuable opinion and feedback. However, we kindly disagree with her/his opinion since this study is part of a larger project and only a subpart of the recorded measures of the project were analyzed for this report. As commonly done in “Brief reports” (Brief reports do only report and analyze some specific measures of interest) and to ensure transparency, we refer the interested reader with the phrase “as described elsewhere” to the previously published study protocol that provides a detailed description of all measures (also of those measures which are not of interest in this study). Based on the specific word limit of “Brief reports”, we feel that we have provided enough information to reproduce our study procedures.

Line 193 – 200 - Too many information. Reduce to minimal necessary to improve clarity.

  • We thank the reviewer for her/his feedback. However, we kindly disagree with her/his opinion. Our reporting of statistical outcomes follows the APA standard where detailed statistical information is required (American Psychological Association, 2020).

Line 213 - The dispersion is usually expressed using the min-max values. Please, consider this to improve clarity.

  • We thank the reviewer for this hint and added the range (min-max) in Table 1 (see line 243 in the revised version of the manuscript) and the Table S1 provided in the supplement.

Line 344/345 - I suggest using a repository (as Mendeley Data) to upload the raw data and these supp. materials.

  • We revised this part as the data availability has already been described elsewhere (see line 408/409) in the revised version of the manuscript). Furthermore, we refrain from using a public repository as MDPI offers the possibility to reposit the supplement directly along with the article.

References

Winblad, B., Palmer, K., Kivipelto, M., Jelic, V., Fratiglioni, L., Wahlund, L.‑O., Nordberg, A., Bäckman, L., Albert, M., Almkvist, O., Arai, H., Basun, H., Blennow, K., Leon, M. de, DeCarli, C., Erkinjuntti, T., Giacobini, E., Graff, C., Hardy, J., . . . Petersen, R. C. (2004). Mild cognitive impairment--beyond controversies, towards a consensus: Report of the International Working Group on Mild Cognitive Impairment. Journal of Internal Medicine, 256(3), 240–246. https://doi.org/10.1111/j.1365-2796.2004.01380.x

This manuscript is a resubmission of an earlier submission. The following is a list of the peer review reports and author responses from that submission.

Round 1

Reviewer 1 Report

The authors studied the relationships between handgrip strength and executive functioning in patients with amnestic and non-amnestic mild cognitive impairment and healthy older adults. 

Some comments.

1) In the introduction you should explain what is "Handgrip strength" and better describe its characteristics. Moreover, it is very important to present the recent studies and the main findings on Handgrip strength, accurately. 

2) The method should be re-designed. Executive functions refer to a wide range of abilities and you can not investigate these skills by means of a single test (in this case the Trail making test, that provides a measure of the attention). This method limits your findings and conclusions. 

3) In table 1, why did you show median and interquartile range? I suggest to present clearly the results reporting mean and standard deviations.  

4) Again, the association between handgrip strength and executive functions (pay attention because you write "executive functions" and "executive function"; you should use "executive functions" only) is not supported and it is should be commented in discussion.  

Reviewer 2 Report

In reference to the study of Herold et al, "A link between handgrip strength and executive functioning: A cross-sectional study in older adults with mild cognitive impairment and healthy controls". The study sets out to answer an interesting question that has clinically relevance concerning a connection between hand-grip strength and cognitive impairment, specifically in MCI and in comparison with healthy control subjects. The study is clearly justified and well written. Unfortunately the study does suffer from several issues of interpretation which are also well summarised by the authors within their own Limitations section. The study is introduced as hypothesis-driven with sound justification, but this is abandoned when attempting to account for their statistical approach. While the correlation values shown in Figure 2 do indeed appear larger than comparisons with HC in the study, there are no significant differences in correlation strength between MCI groups, or between MCI and HC groups. The use of one-sided tests (clearly a directional hypothesis) conflict with a lack of multiple comparisons testing required under the adopted frequentist framework. As such this interesting study feels throughout as if it is stretching for a result. My recommendation would be to estimate effect sizes from this sample and then begin collecting a new dataset with the necessary statistical power to confirm the proposed effect.